# Exploring the transmission mechanism of self discrepancy on perceived academic stress–based on the methods of posting grades ranking

**Shunxu Peng** [1,2], **Feng Yi** [2]*

1 College of Education for the Future, Beijing Normal University, Beijing, China, 2 School of Management, Wuzhou University, Wuzhou, Guangxi, China

* yfcomic@163.com

## Abstract

Academic stress is one of the most significant factors affecting adolescents' mental health. Various methods of posting grades ranking may lead to students to perceive different levels of psychological stress, which in turn impacts the development of their mental health. Using Propensity Score Matching and Ordered Probit methods with educational quality monitoring data of 5405 eighth-grade students from 39 schools in District F, this paper examines the impact of posting grades ranking on students' perceived academic stress in mathematics. It also explores the potential mechanisms and heterogeneous results of this impact, and concludes that: first, self-discrepancy induced by differences on posting grades ranking methods increases students' perceived academic stress, with a higher degree of self- discrepancy leads to greater academic stress to some extent; second, academic stress induced by differences on posting grades ranking methods has a significant impact only on female students, county and rural students, and medium performance group students; third, self-discrepancy contributes to academic stress through the mediating pathway of academic emotions. This study can offer practical insights for recommendations for the implementation of the "Double-Reduction" policy.

## 1. Questions raised

The Overall Plan for Deepening Educational Evaluation Reform in the New Era aims to rectify the unscientific orientation of educational evaluation through reforming the evaluation system. It seeks to overcome the long-standing issue of "Only Five" (Evaluating and selecting talent solely based on scores, promotions, diplomas, theses, and titles.) and emphasizes enhancing outcome evaluation, bolstering process evaluation, exploring value-added evaluation, and improving comprehensive evaluation to establish a system that promotes the all-around development of students. This reform direction has also been advocated abroad, such as in Singapore, where there has

**Data availability statement:** The data used in the research available at https://doi.org/10.6084/m9.figshare.29979691.v1.

**Funding:** The author(s) received no specific funding for this work.

**Competing interests:** The authors have declared that no competing interests exist.

been a commitment to replacing examinations and grades with comprehensive and integrated assessment methods to evaluate students in the past few years⊕.

In order to achieve the goal of education reform, it will be necessary to turn around the trend of exam-oriented education and reduce the excessive academic burden of students. The Opinions on Further Reducing the Burden of Homework on Students in Compulsory Education and the Burden of Out-of-School Training (the "Double Reduction" policy), which was posted in July 2021, explicitly states students' academic burden should be reduced by linking in-school and out-of-school activities. At the same time, the Standards for the Administration of Compulsory Education Schools and the Regulations for the Protection of Minors in Schools have been released to enhance the standardized management of schools. Both documents emphasize that schools are prohibited from disclosing students' test scores and rankings. Currently, ranking based on public examination results remains an important method for academic management in schools and classes, and is a common practice for measuring education quality in many schools. This approach polarizes the assessment function of education, leading many schools to simplify educational assessment by focusing solely on academic achievement rankings. This perpetuates a test-oriented approach, while simultaneously undermining or distorting the fundamental function of educational assessment and objectively increasesing students' academic stress. Effective measures should be implemented to fully achieve the objective of the "Double-Reduction" policy to alleviate students' academic burdens and truly reduce their academic pressure. Regarding the practice of publicly ranking examination results, is it effective in alleviating students' academic stress? The challenge of balancing the reduction of academic pressure through public ranking of examination results and promoting students' academic development deserves further discussion. Based on the current situation of education, this paper will examine whether the manner in which schools publicize student grade rankings can alleviate academic pressure based on the data of monitoring the quality of education for eighth-grade students in a certain place, so as to provide practical experience for the effective implementation of "Double Reduction" policy.

## 2. Literature review and research hypothesis

### 2.1 Academic stress

Stress is a series of biological responses that help organisms adapt to environmental stimuli in order to protect their functional integrity during stressful situation [1].Lazarus and Folkman suggested that stress is a complex response that occurs when individuals encounter unexpected situations that require great effort or exceed their ability to cope, perceiving them as a threat or challenge [2]. Lin et al. identified academic stress as the stress and tension caused by external environmental stimuli or individuals'higher expectations of themselves [3]. Academic stress is a significant stressor that middle school students often experience. It may arise from various sources, including school, family, society, peers, and self. In early adolescence, heightened stress sensitivity and emotional reactivity may make children more susceptible to

life's stresses [4,5]. Excessive academic stress often has a negative impact on students, triggering negative emotions such as anxiety [6] and depression [7]. It also affects sleep quality [8], academic performance [9,10], subjective well-being [11], and school adjustment [12], leading to hypertension, cardiovascular disease [13], and in more severe cases, suicidal ideation [14,15], or even self-injurious and self- suicide behavior. However, Some level of stress has a positive impact on academic performance [16]. Students with high self-esteem will believe that they are capable of coping with academic challenges and demands and will not perceive excessive academic stress [17].Lazarus differentiates between positive challenging stressors and negative hindering stressors that affect performance [18]. Challenging stressors may be accompanied by an increase in attention and performance, whereas hindering stressors may generate anger, leading to deviant or counterproductive behaviors [19].

Academic performance is one of the most important indicators utilized by schools, teachers, and parents to assess students. Students often find themselves competing with their peers to meet the expectations set by educators and guardians. The school, classroom, and family environments in which students are situated can all contribute to the perceived academic pressure experienced by students. In terms of gender groups' perceptions of academic stress, boys and girls do not share the same level of academic stress, and girls tend to experience a higher level of academic stress than boys [20]. Li et al. found that boys experience significantly higher overall academic stress compared to girls [21]. This disparity is attributed to the expectations placed on males by Chinese families and society, which demand more in terms of academic and career achievements. Male students exhibited more problematic behaviors in stressful environments, whereas female students were more likely to experience emotional difficulties [22]. It has also been observed that secondary school students from rural areas experience significantly higher levels of academic stress compared to their urban counterparts [23]. Volitional control is often associated with good academic performance, and High-achieving groups usually have high volitional control, enabling them to maintain focus on learning activities, resist distracting stimuli, and regulate their emotional responses to mitigate anxiety, boredom, and other negative emotions during the learning process [24], which makes the academic stress experienced by high-achieving students is generally lower than that of their low-achieving counterparts. This indicates that there are differences in the perception of academic stress across various factors, including gender, urban versus rural settings, and achievement subgroups.

## 2.2 Academic emotions

Pekrun et al. first explicitly conceptualized academic emotions as emotions directly linked to academic learning, classroom instruction, and academic achievement [25]. The scope of academic emotions is broad, encompassing a variety of emotional experiences related to success and failure in school. They categorized academic emotions by incorporating arousal and dividing them into positive high arousal emotions, positive low arousal emotions, negative high arousal emotions, and negative low arousal emotions. Efklides and Volet identified academic emotions as having three key characteristics: diversity, situationality, and dynamism [26]. Onestudy had found that higher academic stress is associated with students' overall emotion; academic stress is positively correlated with negative academic emotion and negatively correlated with positive emotions [27]. Emotion regulation strategies affect an individual's perception of stress [28]. Another study had found that strategies such as acceptance of emotions, deep breathing, and positive self-expression can help students regulate their negative achievement emotions and improve academic performance [29].

## 2.3 Grades ranking and academic stress

Posting grades ranking is a common practice in most schools and classes in China. The positive impact of exam rankings on enhancing student learning is clear, as exams offer students the chance to face challenges and cultivate a positive outlook on life, along with strong mental resilience skills [30]. Exams serve as an essential educational tool to assess students' comprehension and retention of the material [31]. Individuals possess a strong desire to achieve in both work and study

[32]. Students can fulfill their personal need for achievement by taking exams, which stimulate their motivation to learn and encourage them to enhance their performance. Excessive pressure from competitive examinations can negatively impact students' academic performance by keeping them in a prolonged state of high stress and distraction [33]. Especially in the context of grade ranking in the compulsory education stage, this practice largely distorts the normal function of education, adds significant academic stress to students, and can even lead to serious psychological disorders. In 2015, the online education big data enterprise Afanti utilized the learning behavior big data of 20 million primary and middle school students to release the "National Primary and Middle School Students Academic Stress Survey." The results revealed that Chinese students spend twice as much time on homework as the global average each year, while their daily sleep time is reduced by 1.5 hours. This indicates that Chinese students are facing serious academic stress problems. One study had indicated that the most serious psychological problems among Chinese primary and secondary school students are triggered by the sense of academic stress [34]. Academic stress is the primary source of stress for secondary school students in China, and it is also a significant factor contributing to depression in secondary school students [35]. A white paper on basic education in China released by the New Oriental Education & Technology Group in 2014, over half of the secondary students reported experiencing high academic stress, which mainly stemmed from concerns about grade rankings, parental expectations, self-expectations, pressure to pursue higher education, and competition from classmates. Zhong points out that public ranking is actually against the law of education, causing physical and mental harm to most students, and promoting "examination-oriented," "score-oriented," and "promotion rate supremacy" practices, neglecting the need for quality education to support the all-round development of students [36]. The survey conducted by Liu and Wu on the psychological impact of grade ranking on secondary students revealed that 45.25% of students experienced excessive stress, 35.7% reported weakened self-confidence, and 52.4% indicated that their parents' attitudes were negatively affected by grade ranking [37]. Numerous surveys had also confirmed that grade rankings trigger excessive academic stress among students. However, these studies primarily offer factual descriptions, lack empirical evidence, and do not further analyze the internal mechanism of academic stress caused by grade rankings. In the current context of deepening the reform of the educational assessment system, this paper will investigate the effects of public test score ranking methods on students' academic stress and the mechanisms through which this occurs, based on the theory of self-discrepancy theory.

## 2.4 Self-discrepancy

Higgins proposed the self-discrepancy theory, which distinguishes the self as the actual self, ideal self, and ought self [38]. Different types of self-discrepancy represent various negative psychological conditions associated with psychological discomfort. Discrepancy between one's actual self and ideal self triggers emotions related to rejection, such as disappointment, dissatisfaction, and sadness. Research shows that self-discrepancy is a contributing factor to a variety of mental disorders [39]. Nam et al. stated that differences between the ideal self and the real self were associated with an increase in symptoms of depression and anxiety, as well as the risk of major depressive episodes and generalized anxiety disorder 12 months later. They also found that psychological resilience mediates the relationship between self-discrepancy and mental health [40]. Another study found that self-discrepancy was positively associated with depression and anxiety, and negatively associated with well-being, self-efficacy, tolerance, and resilience [41]. One study from Turkey showed that discrepancy between the actual self and the ideal self was related to depression but not to anxiety. The study also found that psychological resilience moderated the relationship between the discrepancy and anxiety [42]. Self- discrepancy can offer a theoretical framework for understanding how public grade ranking influences students' perceptions of academic stress.

## 2.5 Summary of current research

In the context of Chinese culture, children's education is a significant concern for both teachers and parents, who hold high expectations for their children to receive a quality education in the future. While admitting students based on test scores remains a common practice in schools, the reality is that high-quality educational resources are limited.

Consequently, competing for these opportunities through test scores has become the inevitable choice for most students. The long-established "test-oriented" evaluation system places teachers, parents, and students in an "inward spiral", which contributes to widespread educational anxiety within society. Students are experiencing increased academic pressure, which has drawn attention to the study of academic stress, its sources, and students' responses to it. This focus is particularly relevant in understanding the balance between adolescents'academic development and their mental health. The implementation of the "Double-Reduction" policy is a significant measure by the government aimed at reforming the education evaluation system, eliminating the phenomenon of "inward spiral"and alleviating the anxiety experienced by teachers, parents, and students. Current research has extensively examined the mental health issues faced by adolescents due to academic stress, addressing the promotion of the "Double-Reduction" policy and the reform of public examination result rankings, among other specific practices. However, the existing literature has not provided direct evidence on whether the ranking of public examination results can effectively reduce students' academic stress.

### 2.6 Questions and hypotheses

Based on the summary of the studies presented above, this paper seeks to investigate the following questions: First, what are the characteristics of current manifestations of academic stress among middle school students? Second, does the ranking mode of public examination results influence students' perceptions of academic stress in mathematics? Third, what mechanisms underlie the impact of the ranking mode of public examination results on students' perceptions of academic stress in mathematics?

Based on the issues above, this paper proposes a theoretical model as shown in Fig 1. According to self discrepancy theory, the way students actually rank themselves in the educational setting reflects their actual selves, while their preferred ranking reflects their ideal selves. When there is a mismatch between the actual self and the ideal self, it can lead to feelings of uneasiness and negative emotions [38].The accumulation of these negative emotions can result in psychological stress for students. Therefore, this paper proposes Hypothesis 1: Discrepancy between actual self and ideal self will lead to academic stress. Discrepancy between the actual self and the ideal self can be divided into two cases. The first case is when the actual self is stronger than the ideal self (positive self-discrepancy), and the second case is when the actual self is weaker than the ideal self (negative self-discrepancy). Furthermore, the extent of difference between one's actual self and ideal self may be associated with the intensity of academic stress. Hypothesis 2 proposes that the greater the degree of positive self-discrepancy, the greater the students' perceived academic stress. Conversely, the greater the degree of negative self-discrepancy, the lower the students' perceived academic stress. Combined with self-discrepancy theory, this study proposes Hypothesis 3: self-discrepancy influences students' perceived academic stress through academic emotions.

### 3. Variables and models

The data is derived from a total of 5,405 eighth-grade students in 2021 from 39 schools in District F, which includes 37 public schools and 2 private schools. Of these students, 2,753 (50.93%) were boys.It encompasses the Math Learning

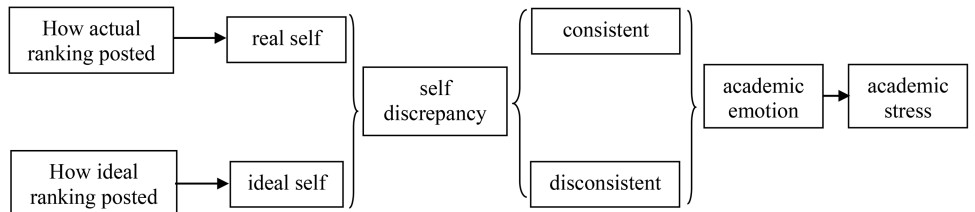

**Fig 1. A theoretical model of the differences in ranking posting methods on academic stress.**

Scale and Physical Fitness Scale, with math scores collected through standardized tests. The scale information includes basic personal and family details, math learning experiences, other instances of math learning, and physical activity. The overall Cronbach's Alpha coefficient for the variables used was 0.782.

### 3.1 Variables

**3.1.1 Dependent variable.** The dependent variable is academic stress related to mathematics. Academic stress refers to the mental burden that students experience during math learning activities, often caused by factors such as excessive homework, grade rankings, and peer competition. Students' self-reports of "no stress at all", "very little stress, feel very relaxed","more stress, often feel a bit overwhelmed" and "too much stress, often feel unable to cope with it" are assigned values of 1, 2, 3, and 4 respectively. Larger values indicate that students perceive more academic stress. The results show that only 6.38% of the students feel no stress at all, 58.94% feel very little stress, 31.49% feel a lot of stress, and 3.19% feel too much stress.

**3.1.2 Independent variable.** The questionnaire asks, "How does your school (class) post rankings of math exams and how would you prefer the rankings to be posted?" The values 1, 2, 3, and 4 were assigned to the statements "Rankings are not announced every time", "The school only tells me and my parents about test scores and rankings but don't announce them", "Rankings are sometimes announced" and "Rankings are announced every time" respectively®. Then, calculate the difference between the actual public grade rankings and the ideal public grade rankings (this difference measures self-discrepancy), which can be greater than 0, equal to 0, or less than 0. When the difference is equal to 0, it indicates self-consistency, while a difference greater than 0 or less than 0 indicates self-inconsistency. Define values greater than 0 or equal to 0 as positive discrepancy groups, and values less than 0 or equal to 0 as negative discrepancy groups. Although the Ministry of Education calls for the elimination of public grade ranking, there may still be significant variations in the way different schools, different classes, and even the same class within the same school publish grade rankings. According to this paper, only 9.49% of the students reported that their school or class does not post grade rankings. 21.35% stated that they or their parents are informed about grades and rankings but they are not posted, while 32.08% mentioned that rankings are sometimes posted by the school or class. Additionally, 37.08% indicated that rankings are posted every time by the school or class.

**3.1.3 Control variables.** Students' academic stress is influenced by a variety of factors, typically including individual, family, and school or classroom factors [43]. Ge Yan noted that learning stress is primarily influenced by several factors, including learning outcomes, the learning process, learning tasks, external expectations placed on students, self-awareness of learning abilities, and self-expectations [44]. Experiencing a high level of academic burden may negatively impact students' academic performance and practices [45].The intrinsic motivation for high achievement [46], along with pressures from parents and teachers such as expectations that explicitly convey the need for adolescents to excel academically [47],which can contribute to increase academic stress among adolescents. It has been confirmed that one of the primary sources of academic stress for many students in China is their parents [48]. It has also been found that after-school physical activity indirectly influences middle school students' academic stress by enhancing their self-confidence, as well as through the chain-mediated effects of increasing both psychological resilience and self-confidence [49]. In addition, there was a significant positive correlation between academic stress and sleep quality [50], While sleep duration is an important dimension in assessing sleep quality, sleep deprivation or drowsiness can lead to issues such as mood disorders [51]. In order to identify the impact of public achievement ranking methods in schools (or classes) on students' perceived academic stress, it is essential to control for relevant variables that influence academic stress.

Individual-level controls include being an only child, personal achievement, interest in learning, self-confidence in learning, test attributions, number of exercises per week, hours of sleep during the week, and hours of sleep on the week-end. Household-level controls encompass place of residence (city, county, rural), father's level of education, and parents' attitudes toward their children's exams. Class-level controls involve whether the math teacher is the teacher in charge of a

class and the degree of class competition. School-level controls account for school fixed effects. Definitions of the various variables are provided in Table 1.

## 3.2 Methods

### 3.2.1 PSM.
Propensity score matching (PSM) is a quasi-experimental method designed to reduce the effects of confounding variables by matching individuals with similar propensity scores in both experimental and control groups,

**Table 1. Definition of the main variables.**

|  | Variables | Description |
|---|---|---|
| Dependent variable | Academic stress | Academic stress in mathematics over the past year |
| Mediating Variable | Academic emotions | 6 sets of phrases describing the emotions of mathematics academics. From difficult to easy, from painful to pleasant, from failure to success, from dull to interesting, from irritating to enjoyable, from obligatory to voluntary. Assign a number from 1 to 7 to each item. Cronbach's alpha coefficient is 0.931 |
| Independent Variable | Self-discrepancy | Difference between the actual school (class) math grade ranking and the preferred grade ranking. Assign a value greater than 0 or equal to 0–1 and 0 in positive self-discrepancy groups. Assign a value less than 0 or equal to 0 to −1 and 0 in negative self-discrepancy groups |
| Control variables | Total score | Math score: the total score is 100 points |
|  | Gender | male=1,female=0 |
|  | Interest in learning | Your level of interest in learning math: very low=1, fairly low=2, average=3, fairly high=4, very high=5 |
|  | Self-confidence in learning | Your level of self-confidence in learning: very low=1, fairly low=2, average=3, high=4, very high=5 |
|  | Examination attribution | Have strong learning ability=1, study hard by yourself=2, have a good teacher=3,the exam content is easy=4, study method is correct=5 |
|  | Frequency of exercises | Number of self-directed exercises per week |
|  | Sleep patterns in general | During the school year, the number of hours of sleep per day is categorized as follows: less than 7 hours=1, 7–8 hours=2, 8–9 hours=3, 9–10 hours=4, and more than 10 hours=5 |
|  | Sleep patterns in weekend | Hours of sleep on weekends or holidays: less than 8 hours=1, 8–9 hours=2, more than 9 hours=3 |
|  | Place of residence | urban=1,county=2,rural=3 |
|  | Only child | yes=0,no=1 |
|  | Father's education level | Elementary school and below=1, middle school=2, high school or junior college=3, college or senior college=4, bachelor's degree=5, graduate school and above=6 |
|  | The way parents approach their children's exams | When I didn't do well on the exam, my parents reacted in the following ways: very angry, assuming I am not smart enough=1;very angry, assuming I don't try hard enough=2;very angry, hitting me and scolding me=3; comforting me, encouraging me, and helping me think of solutions=4, no special reaction=5 |
|  | Teacher in charge of a class | Whether math teacher is a teacher in charge of a class: yes=1, no=0 |
|  | Class Competition | Mean of the class's math scores divided by the standard deviation of the class's math scores |
|  | Score group | Categorize math scores into three groups based on performance in class: the low performance group (bottom 33.3%), the medium performance group (33.3%−66.6%), and the high performance group (top 33.3%) |
|  | School | School dummy variables |

thereby simulating a randomized experimental environment. In this study, three algorithms—nearest neighbor matching, radius matching, and kernel matching—were employed for estimation, providing multiple results and addressing concerns regarding the stability of the estimation outcomes.

Because different schools or classes have adopted various methods of posting grade rankings,and we do not know academic stress of these students if the schools or classes don't adopt a certain method of posting grade rankings. It is challenging to determine the academic stress experienced by students who are subject to various methods of posting grade rankings. Assuming all other factors remain constant, according to self-discrepancy theory, various methods of posting grade rankings influence the actual self and ideal self in both real and ideal environments. The misalignment between the actual self and ideal self leads to negative emotions, which subsequently contribute to academic stress.

The performance of students' self-discrepancy is not random. On one hand, it is influenced by the actual method of posting grade rankings adopted by the school (class). On the other hand, it is also affected by individual expectations of grade rankings, which are shaped by factors such as the individual, family, school, and society. At the same time, these factors also impact students' academic stress. Therefore, the inconsistency group is defined as the difference between the actual method of posting grade rankings adopted by the school (class) and the ideal method, while the consistency group is defined as the similarity between the actual and ideal methods of posting grade rankings. The inconsistency group is considered the treatment group, and the consistency group is considered the control group.

The principle of using PSM estimation is to compare the difference between the mean academic stress levels of the inconsistent group and the consistent group, while ensuring that these two groups are similar in terms of other factors contributing to academic stress. This approach aims to construct a near-randomized environment that maximizes access to the effects of posting grade rankings on students' academic stress.

A counterfactual was constructed using the Propensity Score Matching (PSM) method proposed by Rosenbaum and Rubin [52]. It hypothesizes that for a student, keeping all other characteristics constant, the change in academic stress would be solely due to the way in which their grade ranking is affected by posting. Rosenbaum and Rubin demonstrated that the average treatment effect can be estimated using the following method:

$$ATT = E_{P(X)|D=1} \{ E[Y(1)|D=1, P(byz|X)] - E[Y(0)|D=0, P(byz|X)] \}$$

Where D is a dummy variable representing the experimental or control group; $Y_0$ is the academic stress of students whose actual posting grade ranking adopted by the school (class) is inconsistent with the ideal posting grade ranking, $Y_1$ is the academic stress of students whose actual posting grade ranking adopted by the school (class) is consistent with the ideal posting grade ranking, $X$ is a covariate, and $P(X)$ is the probability of change in self-discrepancy (byz) caused by the experimental group and control group from consistency to inconsistency, while controlling for the covariates of interest. The PSM model satisfies the equilibrium condition assumption and passes the common support test.

**3.2.2 Order-probit.** This paper uses the ordered probit method to analyze the effect of different factors on students' academic stress. The model is as follows:

$$P(y=1|x) = G(\beta_0 + x\beta)$$

The variable $y$ represents a four-valued scale, assigning "1, 2, 3, 4" to "no, small, large, and very large" levels of academic stress. It denotes the factors influencing students' academic stress, which primarily include individual, family, class, and school as previously mentioned. This variable $P$ is continuous and signifies the probability of experiencing academic stress. Further calculate the marginal effect of the independent variable on the dependent variable.

$$\frac{\partial p(x)}{\partial x_j} = g(\beta_0 + x\beta)\beta_j, \quad g(z) = \frac{dG}{dz}(z), g(z) = \exp(z)/[1+\exp(z)]^2$$

*G* is the cumulative distribution function of a continuous random variable, *g* is the probability density function of *G*. This paper estimates the average marginal effect of each independent variable.

## 4. Results and analysis

### 4.1 An overview of perceived academic stress

Table 2 presents an overview of students' perception academic stress in mathematics. Among the full sample, students who did not meet the self-consistency showed a higher percentage of "high" and "great" math academic stress compared to those who did meet the criteria for self-consistency. This trend was particularly pronounced in the positive inconsistency group, while the negative inconsistency group displayed the opposite trend. Thus, Hypothesis 1 was preliminarily tested. Boys have a higher percentage than girls in the "no stress", "little stress" and "great stress" groups, while girls have a higher percentage than boys in the "high stress" group, suggesting potential differences in the perception of academic stress in mathematics between male and female students. The percentage of individuals selecting "high stress and great stress" decreases in the low, medium, and high performance groups, while the percentage choosing "little stress" increases in that sequence, indicating that as achievement levels decrease, students may perceive greater academic stress in mathematics. The percentage of students from county areas reporting "high stress" is higher than that of urban and rural students, while the percentage of county students reporting "great stress" is nearly the same as that of urban students.

### 4.2 Mean effects of self- discrepancy on perceived academic stress

As shown in Table 3, the full sample indicates that self-discrepancy significantly increases students' academic stress in mathematics, supporting Hypothesis 1. The group with a positive discrepancy shows a higher level of students' self-discrepancy, increased academic stress in math, and supports Hypothesis 1. In schools (or classes) that adopt the "posting grade ranking every time" approach, students who expect to be ranked in other ways or not being ranked at all, will experience an increase in academic stress. If schools change their approach from "posting grade rankings every time"

**Table 2. An overview of perceived academic stress.**

| | | Academic stress | | | | |
|---|---|---|---|---|---|---|
| | | No | Little | More | Great | *F* /$\chi^2$ test |
| Total sample | ≠0 | 2.6% | 28.8% | 60.5% | 8.1% | $\chi^2(3)$ =55.17*** |
| | =0 | 3.9% | 34.8% | 57.1% | 4.3% | |
| Positive self-discrepancy group | >0 | 1.9% | 21.8% | 66.4% | 10.0% | $\chi^2(3)$ =147.06*** |
| | =0 | 3.9% | 34.8% | 57.1% | 4.3% | |
| Negative self-discrepancy group | <0 | 4.3% | 44.0% | 47.6% | 4.1% | $\chi^2(3)$ =28.87*** |
| | =0 | 3.9% | 34.8% | 57.1% | 4.3% | |
| Gender | Male | 5.1% | 33.5% | 54.8% | 6.6% | $\chi^2(3)$ =88.29*** |
| | Female | 1.2% | 29.4% | 63.3% | 6.1% | |
| Score group | Low-performance group | 3.38% | 13.63% | 71.42% | 11.57% | $\chi^2(6)$ = 782.9*** |
| | Medium performance group | 1.78% | 31.67% | 61.61% | 4.94% | |
| | High performance group | 4.49% | 51.36% | 42.02% | 2.13% | |
| Residence | Urban | 4.32% | 30.32% | 58.47% | 6.89% | $\chi^2(6)$ =692*** |
| | County | 2.64% | 30.26% | 60.28% | 6.82% | |
| | Rural | 3.24% | 34.24% | 57.22% | 5.30% | |
| Class competition | | 2.82 | 2.93 | 2.89 | 2.87 | *F* =2.92** |
| Academic emotion | | 30.8 | 31.3 | 25.1 | 16.4 | *F* =774.5*** |

**Table 3. The impact of self-discrepancy on academic stress.**

| | | | Experimental group | Control group | ATT | T-value |
|---|---|---|---|---|---|---|
| Total sample | | Pre-matching | 2.71 | 2.64 | 0.08*** | 3.68 |
| | Neighbor matching | Post matching | 2.71 | 2.65 | 0.06*** | 2.49 |
| | Radius matching | Post matching | 2.71 | 2.66 | 0.05*** | 2.40 |
| | Kernel matching | Post matching | 2.71 | 2.66 | 0.06*** | 2.58 |
| Positive self-discrepancy group | | Pre-matching | 2.79 | 2.64 | 0.15*** | 6.94 |
| | Neighbor matching | Post matching | 2.79 | 2.69 | 0.10*** | 3.78 |
| | Radius matching | Post matching | 2.79 | 2.70 | 0.09*** | 3.62 |
| | Kernel matching | Post matching | 2.79 | 2.70 | 0.09*** | 3.70 |
| Negative self-discrepancy group | | Pre-matching | 2.56 | 2.64 | −0.08*** | −2.83 |
| | Neighbor matching | Post matching | 2.57 | 2.56 | 0.00 | 0.01 |
| | Radius matching | Post matching | 2.57 | 2.57 | −0.01 | −0.19 |
| | Kernel matching | Post matching | 2.56 | 2.57 | −0.01 | −0.24 |

Note: (1) The nearest neighbor matching is a 1–4 matching with replacement, and the gauge standard is set to 0.01; the radius matching gauge standard is set to 0.01, and kernel matching is set to default, the same applies here. (2) *** $p < 0.01$, ** $p < 0.05$, * $p < 0.1$, the following is consistent with the standard here.

to "posting grade rankings sometimes" or "only informing oneself or parents of the grades and rankings but not posting them," it will alleviate academic pressure for students who do not expect grade rankings to be posted every time. The negative discrepancy group demonstrates that self-discrepancy does not cause significant negative stress for students. Instead, it can be interpreted as having a positive psychological comfort effect. In other words, if a student consistently aims to achieve top grades ranking(posting grade ranking every time), they may not experience a decrease in academic stress when the method of posting grade rankings is altered.

As shown in Table 4, both the positive discrepancy group and the negative discrepancy group contain three cases(There are insufficient samples for difference of −3; therefore, no results are available.). In the positive discrepancy group, a greater difference in self-discrepancy leads to higher overall academic stress. It can be understood that when all other conditions are constant, a student who prefers "not to post grade rankings every time" in schools will experience a gradual increase in academic stress compared to students who have the options of "the school only tells me and my parents about the test scores and rankings, but does not post them", "sometimes post grade rankings", and "posting grade ranking every time". For the negative discrepancy group, the change in self-discrepancy does not lead to academic stress as a whole, meaning there is no positive psychological compensation effect. Hypothesis 2 was partially tested.

### 4.3 Mechanism analysis of self- discrepancy on perceived academic stress

Based on the theoretical model in Fig 1, this study aims to examine how self-discrepancy, as represented by different methods of posting grade rankings, influences students' academic stress in mathematics. As depicted in Table 5, the coefficient of self-discrepancy in Model 1 is significantly positive, suggesting that self-discrepancy contributes to academic stress. The coefficient for self-discrepancy in Model 2 is significantly negative, suggesting that self-discrepancy contributes to more negative academic emotions. The coefficients for both self-discrepancy and academic emotion are significant in Model 3, indicating that academic emotion serves as a mediator in the process of self-discrepancy leading to academic stress. The marginal effects reveal that the impact of self-discrepancy and academic emotion on academic stress varies in two situations. On one hand, for students experiencing high levels

**Table 4. The effect of various levels of difference in self-discrepancy on academic stress.**

| | | | | Experimental group | Control group | ATT | T-value |
|---|---|---|---|---|---|---|---|
| Positive self-discrepancy group | Difference=1 | | Pre-matching | 2.73 | 2.64 | 0.09*** | 3.66 |
| | | Neighbor matching | Post matching | 2.72 | 2.65 | 0.07** | 2.35 |
| | | Radius matching | Post matching | 2.73 | 2.64 | 0.09*** | 3.40 |
| | | Kernel matching | Post matching | 2.73 | 2.65 | 0.07*** | 2.80 |
| | Difference=2 | | Pre-matching | 2.85 | 2.64 | 0.21*** | 7.67 |
| | | Neighbor matching | Post matching | 2.84 | 2.79 | 0.05 | 1.41 |
| | | Radius matching | Post matching | 2.84 | 2.78 | 0.07* | 1.94 |
| | | Kernel matching | Post matching | 2.85 | 2.77 | 0.08** | 2.44 |
| | Difference=3 | | Pre-matching | 3.06 | 2.60 | 0.46*** | 6.88 |
| | | Neighbor matching | Post matching | 3.06 | 2.78 | 0.28*** | 3.02 |
| | | Radius matching | Post matching | 3.06 | 2.78 | 0.29*** | 3.02 |
| | | Kernel matching | Post matching | 3.06 | 2.85 | 0.21** | 2.35 |
| Negative self-discrepancy group | Difference=−1 | | Pre-matching | 2.57 | 2.64 | −0.07** | −2.56 |
| | | Neighbor matching | Post matching | 2.57 | 2.57 | 0.01 | 0.18 |
| | | Radius matching | Post matching | 2.57 | 2.56 | 0.01 | 0.34 |
| | | Kernel matching | Post matching | 2.57 | 2.57 | 0.00 | 0.13 |
| | Difference=−2 | | Pre-matching | 2.51 | 2.67 | −0.15** | −2.14 |
| | | Neighbor matching | Post matching | 2.51 | 2.69 | −0.17* | −1.76 |
| | | Radius matching | Post matching | 2.51 | 2.68 | −0.16* | −1.73 |
| | | Kernel matching | Post matching | 2.51 | 2.63 | −0.12 | −1.35 |

**Table 5. The test result of self discrepancy affecting process of academic stress (positive stress group).**

| | Oprobit model | | | Marginal effect of academic stress | | | |
|---|---|---|---|---|---|---|---|
| | Model1 | Model2 | Model3 | | | | |
| Dependent variable | Academic stress | Academic emotion | Academic stress | No | Little | More | Great |
| Independent variable | | | | | | | |
| Self-discrepancy | 0.160*** | −0.080* | 0.134*** | −0.006*** | −0.030*** | 0.023*** | 0.134*** |
| | (0.045) | (0.042) | (0.046) | (0.002) | (0.010) | (0.008) | (0.005) |
| Mediating variable | | | | | | | |
| Academic emotion | | | −0.376*** | 0.017*** | 0.085*** | −0.064*** | −0.038*** |
| | | | (0.030) | (0.002) | (0.006) | (0.005) | (0.003) |
| Log likelihood | −2577.01 | −3698.43 | −2360.62 | | | | |
| Number | 3252 | 3258 | 3244 | 3244 | 3244 | 3244 | 3244 |
| Pseudo R² | 0.132 | 0.171 | 0.202 | | | | |

of academic stress, self-discrepancy exacerbates their stress by amplifying negative academic emotions. On the other hand, for students with low or no academic stress, self-discrepancy enhances positive academic emotions and reduces their academic stress. Meanwhile, when the dependent variable is a categorical variable and the mediator variable is academic emotion as a continuous variable, the generalized structural equation modeling (gsem) method can be used to test the mediating effect [53]. As indicated in Table 6, the test results align with those in Table 5. The total effect is 0.243, and the indirect effect is 0.079, both of which are significant (p < 0.001). The mediating effect accounts for 32.5% of the total effect, verifying Hypothesis 3.

**Table 6. Results of the gsem model test for mediating effects(positive stress group).**

| Variables | Academic emotion | Academic stress |
|---|---|---|
| Academic emotions | | −0.268*** |
| | | (0.008) |
| Self-discrepancy | −0.295*** | 0.085*** |
| | (0.035) | (0.018) |
| Constant | 2.52*** | 3.30*** |
| | (0.028) | (0.024) |
| Number | 4131 | 4120 |

### 4.4 Heterogeneity analysis of self- discrepancy on perceived academic stress

**4.4.1 Gender difference.** As shown in Table 7, overall, the results of the gender sub-sample in the positive self-discrepancy group indicate a gender difference in the effect of self-discrepancy on students' academic stress. Self-discrepancy does not have a significant effect on academic stress for males, but it does cause significant academic stress for females. A related study also found that female students experienced more academic stress than male students [54]. For this reason, it has been pointed out that there is a close relationship between students' core self-evaluation and perceived academic stress [55].Since boys tend to be endowed with more gender role expectations from their families, they develop a more positive core self-evaluation as they grow up [56].This positive trait will help them cope with academic stress. Boys and girls differ in their interpersonal communication patterns. Girls tend to focus more on relationships and emotional experiences, leading to interactions that are generally narrower and more focused [57].However, this approach may not be conducive for them to alleviate academic stress. Men and women respond differently to stress, with men tending to choose more competitive fields and women choosing less competitive fields [58]. Additionally, women show an aversion to risk[6 59], which makes them more susceptible to stress. At the same time, girls demonstrate greater sensitivity and empathy towards their surroundings. They are also more likely to experience stress in competitive environments, which can lead to poorer performance compared to boys under similar conditions. Overall, a negative self-discrepancy does not have a positive psychological compensatory effect on either boys or girls.

**4.4.2 Urban-rural differences.** The urban and rural sub-sample in table 8 indicates that self-discrepancy significantly increases academic stress among urban students, while it has no significant effect on the academic stress of county and rural students. On one hand, urban and rural students experience different family environments. Urban parents tend to have higher levels of education compared to their counterparts in counties and rural areas. Parents with higher education

**Table 7. Gender differences in self-discrepancy affecting academic stress.**

| | | | Positive self-discrepancy group | | | | Negative self-discrepancy group | | | |
|---|---|---|---|---|---|---|---|---|---|---|
| | | | Experimen-tal group | Control group | ATT | T-value | Experimental group | Control group | ATT | T-value |
| Male | | Pre-matching | 2.75 | 2.57 | 0.18*** | 5.25 | 2.50 | 2.57 | −0.07* | −1.71 |
| | Neighbor matching | Post matching | 2.74 | 2.70 | 0.04 | 1.07 | 2.50 | 2.52 | −0.02 | −0.32 |
| | Radius matching | Post matching | 2.74 | 2.69 | 0.05 | 1.31 | 2.50 | 2.51 | −0.00 | −0.09 |
| | Kernel matching | Post matching | 2.75 | 2.67 | 0.09** | 2.11 | 2.50 | 2.51 | −0.00 | −0.09 |
| Female | | Pre-matching | 2.82 | 2.70 | 0.12*** | 4.47 | 2.63 | 2.70 | −0.06* | −1.96 |
| | Neighbor matching | Post matching | 2.81 | 2.76 | 0.05* | 1.60 | 2.63 | 2.67 | −0.04 | −0.95 |
| | Radius matching | Post matching | 2.81 | 2.75 | 0.06* | 1.96 | 2.63 | 2.65 | −0.02 | −0.44 |
| | Kernel matching | Post matching | 2.81 | 2.75 | 0.06** | 2.14 | 2.63 | 2.65 | −0.02 | −0.48 |

**Table 8. Rural-Urban differences in self-discrepancy affecting academic stress.**

| | | | Positive self-discrepancy group | | | | Negative self-discrepancy group | | | |
|---|---|---|---|---|---|---|---|---|---|---|
| | | | Experimental group | Control group | ATT | T-value | Experimental group | Control group | ATT | T-value |
| Urban | | Pre-matching | 2.80 | 2.61 | 0.19*** | 4.04 | 2.54 | 2.62 | −0.07 | −1.20 |
| | Neighbor matching | Post matching | 2.79 | 2.67 | 0.12* | 1.92 | 2.55 | 2.46 | 0.09 | 1.20 |
| | Radius matching | Post matching | 2.79 | 2.67 | 0.12* | 1.94 | 2.55 | 2.45 | 0.09 | 1.28 |
| | Kernel matching | Post matching | 2.79 | 2.68 | 0.11* | 1.94 | 2.55 | 2.47 | 0.08 | 1.11 |
| County | | Pre-matching | 2.81 | 2.67 | 0.15*** | 4.61 | 2.59 | 2.66 | −0.07* | −1.97 |
| | Neighbor matching | Post matching | 2.80 | 2.75 | 0.05 | 1.41 | 2.59 | 2.61 | −0.02 | −0.46 |
| | Radius matching | Post matching | 2.80 | 2.76 | 0.04 | 1.23 | 2.59 | 2.61 | −0.02 | −0.36 |
| | Kernel matching | Post matching | 2.80 | 2.74 | 0.06 | 1.68 | 2.59 | 2.60 | −0.00 | −0.08 |
| Rural | | Pre-matching | 2.74 | 2.62 | 0.12*** | 3.19 | 2.54 | 2.62 | −0.08* | −1.71 |
| | Neighbor matching | Post matching | 2.72 | 2.68 | 0.03 | 0.65 | 2.54 | 2.56 | −0.02 | −0.37 |
| | Radius matching | Post matching | 2.72 | 2.69 | 0.02 | 0.53 | 2.54 | 2.56 | −0.02 | −0.35 |
| | Kernel matching | Post matching | 2.72 | 2.69 | 0.03 | 0.68 | 2.54 | 2.56 | −0.02 | −0.36 |

levels often have higher expectations for their children's education, which may lead to academic burden and psychological stress for their children. On the other hand, urban and rural students experience different school environments. The teaching standards and requirements in urban schools are higher compared to those in county and rural schools. Urban students also demonstrate greater dedication to their studies than their county and rural counterparts, leading to heightened competition among peers. Consequently, this high level of engagement inevitably places an excessive psychological burden on them. The data in this paper also shows that urban students have more negative emotions towards math academics compared to students from counties and rural areas, which may result in greater academic stress for urban students.

**4.4.3 Score group difference.** A subsample of various performance groups in Table 9 indicates that only students in low and medium performance groups experience significant academic stress due to self-discrepancy; students in high-performance groups do not experience this stress. Overall, there is a gradual decrease in academic stress and a steady rise in negative academic emotions among students in the low, medium, and high performance groups. Under the current education system, middle school students encounter a screening environment similar to that of high school. There is a lack of emphasis on the concept of "equality between general education and vocational education" The competitive stress brought about by the categorization of "middle school examination" is no less than that of the "college entrance examination" Low- and medium-performing students are often more susceptible to the impact of this policy environment and experience more serious academic stress. At present, despite the implementation of the "Double Reduction" policy in the education evaluation system, the practice of assessing schools and students based on their performance, which has been established by schools, teachers, parents, and society, remains deeply ingrained. Emphasis is still placed on students with outstanding performance rather than on those with low or moderate performance. This may make them more likely to form negative self-evaluations, display more negative social emotions, and perceive more academic stress. In the group with negative discrepancies, self-discrepancy does not significantly contribute to academic stress for students in various performance groups.

### 4.5 Research limitations

Although this paper provides partial evidence that the manner in which schools (or classes) publish grade rankings affects students' academic stress, it also identifies that the method of grade publication influences academic stress through the

**Table 9. Performance subgroup differences in self-discrepancy affecting academic stress.**

| | | | Positive self-discrepancy group | | | | Negative self-discrepancy group | | | |
|---|---|---|---|---|---|---|---|---|---|---|
| | | | Experimental group | Control group | ATT | T-value | Experimental group | Control group | ATT | T-value |
| Low performance group | | Pre-matching | 2.99 | 2.85 | 0.14*** | 4.00 | 2.85 | 2.85 | −0.0 | −0.03 |
| | Neighbor matching | Post matching | 2.98 | 2.90 | 0.08* | 1.91 | 2.83 | 2.83 | 0.00 | 0.03 |
| | Radius matching | Post matching | 2.98 | 2.88 | 0.10** | 2.36 | 2.83 | 2.82 | 0.01 | 0.10 |
| | Kernel matching | Post matching | 2.98 | 2.88 | 0.11** | 2.56 | 2.84 | 2.81 | 0.03 | 0.54 |
| Medium performance group | | Pre-matching | 2.79 | 2.65 | 0.14*** | 3.95 | 2.60 | 2.65 | −0.05 | −1.19 |
| | Neighbor matching | Post matching | 2.78 | 2.69 | 0.09* | 1.96 | 2.59 | 2.65 | −0.06 | −1.13 |
| | Radius matching | Post matching | 2.78 | 2.70 | 0.08* | 1.85 | 2.59 | 2.65 | −0.06 | −1.15 |
| | Kernel matching | Post matching | 2.78 | 2.70 | 0.09** | 2.28 | 2.59 | 2.62 | −0.02 | −0.47 |
| High performance group | | Pre-matching | 2.49 | 2.43 | 0.06 | 1.42 | 2.39 | 2.43 | −0.03 | −0.84 |
| | Neighbor matching | Post matching | 2.48 | 2.51 | −0.02 | −0.45 | 2.40 | 2.42 | −0.03 | −0.56 |
| | Radius matching | Post matching | 2.48 | 2.52 | −0.04 | −0.84 | 2.40 | 2.41 | −0.02 | −0.39 |
| | Kernel matching | Post matching | 2.48 | 2.51 | −0.03 | −0.59 | 2.39 | 2.41 | −0.02 | −0.34 |

mediating pathway of academic emotions. However, the study has several limitations. First, this paper utilized one-year cross-sectional data from District F. Although propensity score matching was employed to identify the effects on students' perceived academic stress due to changes in the way grades are announced, it is challenging to assert a definitive causal relationship between these variables. Second, the theoretical foundation of this paper is somewhat weak. While it articulates the core logic of the theory of self-discrepancy, and captures some relevant information of self-discrepancy, it remains distant from the concept of self-discrepancy as understood by mainstream psychology. Additionally, the data do not provide sufficient information to measure self-discrepancy, which limits the validation of the role of self-discrepancy theory in this study. Third, the analysis of the mechanism by which self-discrepancy affects students' perceptions of academic stress may lack persuasiveness, and there is an absence of a convincing theoretical framework in the current research. Consequently, this study engages in bold explorations with the aim of enriching the content of this field of research.

## 5. Summary and discussion

In order to achieve the goal of comprehensive development for students, it is necessary to deepen the reform of the education evaluation system and break down the barriers of exam-oriented education. The "Double Reduction" policy and the strengthening of standardized school management are intended to reduce excessive academic stress on students. By clearly defining the responsibilities of families and schools, the aim is to bring education back to its roots and lay the foundation for shifting away from an examination-oriented approach. Academic stress can stem from various sources. This paper demonstrates that students believe the highest percentage of academic stress in mathematics comes from intense competition among classmates (39.6%), followed by social stress, higher education, job search, and other issues (26%)[③]. Generally speaking, students' academic stress depends on self-perception, and everyone perceives stress differently. Whether the abolition of posting grade rankings can alleviate students' academic stress cannot be conclusively determined. This paper indirectly assesses the impact of changing the method of posting grade rankings on students' academic stress perceptions by examining the self-discrepancy between the actual and ideal ways of announcing rankings for students. First, the study identifies that differences in the way students perceive grade rankings can lead to self-discrepancy and increased academic stress in mathematics. Furthermore, a higher degree of self-discrepancy is associated with greater academic stress to some extent. Second, academic stress caused by differences in grade ranking has

a significant effect only on female, county, rural, and middle-performing group students. Third, self-discrepancy resulting from variations in grade ranking can lead to academic stress through emotional responses to self discrepancy.

In the past, many schools have adopted the practice of evaluating teachers and students based on grade rankings. This practice undoubtedly reinforces a sense of stress among students who are more sensitive to it. For instance, students who prefer not to have rankings every time are more likely to experience additional academic stress in reality. The abolition of public grade rankings is likely to be more beneficial for students who are particularly sensitive to stress, while it may not positively impact those who are less affected by academic pressure. However, this practice carries potential risks. On one hand, it may diminish the motivational function of scores and rankings, leading some highly utilitarian students to experience less self-investment in learning and negatively affect their academic performance; On the other hand, the pedagogical assessment function of scores and rankings may also be weakened, making it difficult for teachers to accurately assess their students' learning situations in a timely manner. This lack of insight could hinder their ability to implement effective academic interventions. Moreover, eliminating the posting of grade rankings may cause anxiety among parents who have a high levels of control and lack insight into their children's performance.

In fact, both schools and teachers, as well as parents, have a relatively clear understanding of the current education system. They all make personal choices and follow a path based on the system. For schools and teachers, it is challenging and expensive to shift their dependence on traditional assessment methods to comply with the policy. For many parents, schooling holds significant memories of shaping their future. They aspire for their children to excel and recognize that success often hinges on competing for a limited number of opportunities. Students often experience stress from their academic workload and personal life, and face intense competition from their peers. Even when schools and parents aim to alleviate the psychological burden on students through conventional methods, they are still impacted by the underlying emphasis on competition and the traditional societal pressure to "get ahead." As a result, they are compelled to prioritize short-term achievements over their children's long-term well-being, leading to a pervasive trend of educational involution. Currently, education is often seen primarily for its instrumental value, with schools, teachers, parents, and students viewing education as a means for progress. As a result, it is challenging for students to internalize education as a pathway to rational self-development. This is a common observation in universities, where many students experience a sudden lack of constraints from their families and teachers, leading to a loss of interest in learning. They may feel confused for a long time and eventually choose to "lie flat."

Actually, even if grade rankings are abolished, students may still be concerned about their grades and make comparisons with their classmates by making private inquiries. In that case, students may assign themselves rankings in their minds and gradually shift the focus from public competition among classmates to competition between students and themselves. This shift may help students develop self-awareness and stimulate motivation to challenge themselves. Moreover, the goal of the "Double Reduction" policy is not to reduce normal academic stress, but to alleviate the unnecessary academic stress caused by unreasonable practices, such as excessive homework and extracurricular tutoring. On the contrary, the quality of teaching and learning should be improved by enhancing teaching efficiency, so that students can achieve self-development while managing reasonable academic stress.

Based on this, the decision to abolish grade rankings has its rationale. However, the implementation of this policy may also be complex. Therefore, in the process of comprehensively promoting the reform of the education evaluation system and implementing "Double Reduction" policy, it is necessary to adopt a "school-based policy" (Use methods that align with the local school circumstances. This is an idiom in China.) approach rather than implementing a one-size-fits-all approach to policy implementation. For instance, we can learn from the United States, which publishes test results in the form of report card and provides only average grade scores. This approach helps alleviate students' anxiety, as it reduces the pressure associated with test rankings.It is more desirable to adopt a strategy of gradual and phased advancement in order to establish a foundation for continuous reform. At the same time, it is necessary to respect the laws of education and school operation, fully assess the impact of the policy on different groups, activate the initiative of schools and

teachers, involve parents in the school, and establish a positive interactive relationship with parents to collectively promote the development of students.

① https://www.moe.gov.sg/education-in-sg/our-students

② Here is the frequency of students' self-reported "non-disclosure of grade ranking". Given potential variations in how schools (or classes) implement public grade ranking, and the possibility of different answers from students within a school (or class), it is not feasible to definitively establish the actual public grade ranking within a school (or class). Thus, based on the information that "Rankings are not announced every time", "The school only informs me and my parents about the test scores and rankings, but does not publicly announce them", "Sometimes the rankings are announced" and "Ranking is announced every time" values of 1, 2, 3, and 4 were assigned. These values were then weighted within the class based on the number of students who preferred a specific method of public grade ranking, and rounded up to determine the actual method of public grade ranking adopted by the class.

③ Other sources of stress include: demanding too much of myself (17%), my parents demanding a lot from me (12.3%), and the demands of schools and teachers (5.2%).

## Acknowledgments

The authors would like to thank all of the people who participated in the studies.

## Author contributions

**Conceptualization:** Shunxu Peng.

**Formal analysis:** Shunxu Peng.

**Methodology:** Shunxu Peng.

**Supervision:** Feng Yi.

**Writing – original draft:** Shunxu Peng.

**Writing – review & editing:** Feng Yi.

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
