## [Decision Letter · Decision Letter 0]

23 Jan 2025

PONE-D-24-06473Exploring the Transmission Mechanism of Self discrepancy on Perceived Academic Stress--Based on the methods of posting grades rankingPLOS ONE

Dear Dr. Peng,

Thank you for submitting your manuscript to PLOS ONE. After careful consideration, we feel that it has merit but does not fully meet PLOS ONE’s publication criteria as it currently stands. Therefore, we invite you to submit a revised version of the manuscript that addresses the points raised during the review process.

**ACADEMIC EDITOR:** Dear Authors, Thank you once again for submitting your research to PLOS ONE.The manuscript deals with an important and timely topic that can contribute to the literature in multiple ways.Having carefully reviewed your manuscript and having consulted the reviewer's feedback, I have decided that the manuscript can be published should you be prepared to incorporate minor revisions. I outline below my requested revisions and hope that you will be willing to complete these necessary revisions as soon as possible in order to facilitate a timely decision on your revised manuscript. Looking forward to receiving the revised version in due course!==============================

We look forward to receiving your revised manuscript.

Kind regards,

Ioannis G. Katsantonis, PhD, MPhil

Academic Editor

PLOS ONE

Journal Requirements:

2. In the online submission form, you indicated that your data is available only on request from a third party. Please note that your Data Availability Statement is currently missing contact details for the third party, such as an email address or a link to where data requests can be made. Please update your statement with the missing information.

3. Please ensure that you include a title page within your main document. You should list all authors and all affiliations as per our author instructions and clearly indicate the corresponding author.

6. Please include your tables as part of your main manuscript and remove the individual files. Please note that supplementary tables (should remain/ be uploaded) as separate "supporting information" files

**Additional Editor Comments:**

This is an interesting study utilising a large sample size and advanced analytic methods, such as propensity score matching and ordered probit modelling. I find it impressive that the data are representative and that advanced analytic techniques have been used to analyse the data. However, a couple of outstanding issues should be resolved before the manuscript can be published. Please find below my feedback.

1. The abstract is unstructured and should be revised to include basic information such as the sample size and the proportion of males vs females.

2. The introductory section provides an interesting context by introducing the double-reduction policy and its implications. Yet, the key outcomes of the analyses are not mentioned at all. I recommend bringing the key outcomes into focus as well.

3. In section 3 (Grades Ranking and Academic Stress), the first paragraph requires further citations to support the uncited arguments.

4. Some references inside the manuscript do not follow the PLOS referencing style. Additionally, I recommend formatting the manuscript closely following the journal templates and guidelines.

5. The first paragraph in page 13 should constitute the 'present study' subsection of the literature review. In addition to these hypotheses, I recommend adding research questions. Also, please format the hypotheses using bullet points for a more convenient access to these.

6. The authors mention validity and reliability. Reliability has been assessed using the Cronbach's alpha coefficient, I presume, but it is unclear whether factor analysis has been used to ensure the validity of the measures (multi-item measures).

7. I would like to see a better justification of the discrepancy measure. Perhaps, the authors could consult some literature and present such evidence backing their discrepancy measure.

8. Some empirical literature should be cited to justify the selected control variables.

9. Inside the results' section, the authors present heterogeneity analyses by gender, urban status, and score group differences. These analyses are not clearly substantiated by relevant theory or evidence inside the literature review's section. I would urge the authors to add a short subsection describing how these factors can induce heterogeneity.

10. Before the conclusion of the article, please add a comprehensive limitations' section.

11. In the PSM method, please explain more the different matching algorithms implemented and their strengths. Also, try to outline the benefits of using this quasi-experimental method.

12. Furthermore, the tables have been presented as pictures/images in ./tiff format. However, it is the PLOS principle that tables should be presented as editable tables. Please revise accordingly.

Reviewers' comments:

Reviewer's Responses to Questions

**Comments to the Author**

1. Is the manuscript technically sound, and do the data support the conclusions?

Reviewer #1: Yes

2. Has the statistical analysis been performed appropriately and rigorously? 

Reviewer #1: Yes

3. Have the authors made all data underlying the findings in their manuscript fully available?

Reviewer #1: No

4. Is the manuscript presented in an intelligible fashion and written in standard English?

Reviewer #1: Yes

5. Review Comments to the Author

Reviewer #1: Dear Authors,

Thank you for such an interesting and important article!

A section with the limitations of the study should be added to the article. All the necessary aspects that may arise regarding this study need to be detailed.

The illustrations that are presented in the article are of poor quality.

6. PLOS authors have the option to publish the peer review history of their article (what does this mean? ). If published, this will include your full peer review and any attached files.

**Do you want your identity to be public for this peer review?** For information about this choice, including consent withdrawal, please see our Privacy Policy .

Reviewer #1: No

---

## [Author Response · Author response to Decision Letter 1]

29 Mar 2025

1.Response to academic editors:We have rechecked and revised the original manuscript as requested by the academic editor.

2.We have submitted the analyzed data in the supplementary material.

3.Response to additional editor

(1) The abstract is unstructured and should be revised to include basic information such as the sample size and the proportion of males vs females.

We have revised the abstract. Necessary information related to the sample was added and a statement of the significance of the study was added. Considering the brevity of the abstract, more information was not added. Also ,in the place of the variables and methods section ,basic information such as gender share was added.

(2)The introductory section provides an interesting context by introducing the double-reduction policy and its implications. Yet, the key outcomes of the analyses are not mentioned at all. I recommend bringing the key outcomes into focus as well.

Many thanks to the editors for the pertinent comments. We have tried to explain this as follows. First, in the summary and discussion section, it is mentioned that: This study found that it is the self-discrepancy caused by changes in the way test scores are ranked that contributes to individuals' perceptions of academic stress, and so responds to the inaccuracy of the claim that current policy requires that the elimination of grade ranking reduces students' academic stress. Second, the discussion adds potential risks that may result from the elimination of public grade ranking practices, including risks to students, teachers, and parents. Third, it discusses the dilemmas faced in the implementation of the current policy of promoting the reform of the education evaluation system. Fourth, it discusses the possible future direction of the policy and gives some suggestions. Therefore, in the summary and discussion section, this paper closely follows the findings of this study and gives necessary responses to the current practice of promoting the reform of the education evaluation system and the “Double-Reduction” policy.

(3) In section 3 (Grades Ranking and Academic Stress), the first paragraph requires further citations to support the uncited arguments.

We agree with the comments made by the editors. The first paragraph of Part III in Literature review and research hypothesis(Grades Ranking and Academic Stress) has been supplemented with relevant literature on the relationship between grades ranking and academic stress, and the relevant formulation has been adjusted.

(4)Some references inside the manuscript do not follow the PLOS referencing style. Additionally, I recommend formatting the manuscript closely following the journal templates and guidelines.

We have made changes following the requirements of the journal literature format.

(5)The first paragraph in page 13 should constitute the 'present study' subsection of the literature review. In addition to these hypotheses, I recommend adding research questions. Also, please format the hypotheses using bullet points for a more convenient access to these.

We agree with the changes proposed by the editors. The literature review was first summarized independently, and then the relevant statement of the research question was added, which was not introduced separately, considering the unity of the research hypothesis and research question. We do not know if this treatment is reasonable, and we will continue to improve it if the editor has additional comments.

(6) The authors mention validity and reliability. Reliability has been assessed using the Cronbach's alpha coefficient, I presume, but it is unclear whether factor analysis has been used to ensure the validity of the measures (multi-item measures).

We didn't present it clearly in the manuscript, Cronbach's alpha coefficient was used for reliability and factor analysis was used for validity analysis. In the operation process, for the math scale, the relevant variables involved in the basic personal and family information were excluded, and the math learning situation, other relevant situations (a total of 38 questions, which are related to the extracurricular learning of math, The degree of fear associated with learning mathematics, etc.) and the math test scores were added to calculate the Cronbach's alpha coefficient, and the principal component analysis in Spss was used to extract the common factor to calculate the KMO value and determine the validity of the Bartlett's alpha coefficient. In the original manuscript, the reliability is the Cronbach's alpha coefficient and the validity is the KMO value. The reliability and validity of the sports scales were calculated in the same way. Finally, the overall reliability and validity were calculated by combining the math and sports scales together. After referring to the relevant literature, the original manuscript was modified to report only the Cronbach's alpha coefficients. Variables used in the calculations included: academic stress, how ideal grades are posted, how actual grades are posted, Academic emotions, interest in learning, Self-confidence in learning, length of sleep on weekends, length of sleep on weekdays, Frequency of exercises, parents' approach to their children's grades, level of class competition, positive attributions for examination, and Score group.

(7)I would like to see a better justification of the discrepancy measure. Perhaps, the authors could consult some literature and present such evidence backing their discrepancy measure.

Higgins (1987) integrated different aspects of self-representation to propose a theory of self-discrepancy. He argued that two cognitive dimensions—domain of the self and standpoints on the self—underlie the different representations of the self. The domain of the self includes (1) actual self, (2) ideal self, and (3) ought self, while standpoints on the self encompass (1) one's own personal standpoint and (2) the standpoint of significant others. I agree with the editors that measuring self-discrepancy is a problem that cannot be avoided.

Higgins et al. (1985) developed a questionnaire to measure the level of self-discrepancy by asking participants to select appropriate adjectives for different domains of the self: Actual, Ideal, and Ought. They evaluated the degree to which each adjective reflected their characteristics using a 4-point Likert scale. The degree of consistency across these domains was assessed by identifying the differences in how pairs of adjectives for the actual and ideal selves were rated as a “match, some match, or mismatch” according to a certain set of rules.

This way of measuring self-discrepancy modified by Hardin (2002), assesses the degree of perceived discrepancy between actual self, ideal self, and ought self. Participants are asked to write five characteristics that correspond to the self-concept, referencing a list of 100 trait-reflective words. They then rate each of the listed characteristics on a 5-point scale. This classic measure of self-discrepancy essentially reflects subjects being in a present state (actual self) versus a desired state (ideal self). Ideally, students would be themselves to report that their perceptions of whether or not they have publicized grade rankings gained representations of their current and ideal selves (by listing a few adjectives reflecting their feelings about their school or class's publicized grade rankings, such as disgusted, satisfied, etc., and rated on a 5-point scale), this study is not a self-report measure of the actual and ideal selves. Instead, it examines the differences in the actual public grade ranking approach adopted by the school (or class) to reflect the current state of the self. It also considers student-reported preferences for a specific public grade approach to represent the ideal self. This method captures the actual and ideal selves to some extent. While this study does not strictly adhere to the traditional psychological research paradigm, it borrows its core logic and adopts a different approach to measure the actual and ideal selves.

(8)Some empirical literature should be cited to justify the selected control variables.

Relevant literature has been added to the text of the first paragraph on control variables in the Variables and Methods section to justify the choice of the following control variables.

(9)Inside the results' section, the authors present heterogeneity analyses by gender, urban status, and score group differences. These analyses are not clearly substantiated by relevant theory or evidence inside the literature review's section. I would urge the authors to add a short subsection describing how these factors can induce heterogeneity.

We agree with the editors and have added a paragraph to the Academic Stress section of the Literature Review describing the literature on the differential manifestation of academic stress in different groups.

(10) Before the conclusion of the article, please add a comprehensive limitations' section.

We agree with the editors and have added a comprehensive limitations section in the appropriate place.

(11)In the PSM method, please explain more the different matching algorithms implemented and their strengths. Also, try to outline the benefits of using this quasi-experimental method.

We have added a statement in the Methods section that introduces the respective advantages of different matching algorithms and briefly summarizes the advantages of using the proposed experimental method. And here we make the following elaboration on the respective advantages of each matching algorithms, (1) Near-neighbor matching, whose operation is based on matching operates by pairing individuals with the closest propensity scores from both the experimental and control groups. This method ensures that all individuals in the experimental group are matched, thereby preventing sample wastage, which is particularly beneficial when the experimental group has a small sample size. However, if the disparity between the propensity score distributions of the experimental group and the control group is too large, it may lead to poor matching quality, thus affecting the estimation accuracy. In this study, different approaches such as 1-to-1,1-to-many, etc. were tried and caliper standards (0.01) were set, and similar results were obtained, and finally the results of the 1-to-4 matching algorithm with put-back were used. (2) Radius matching, which operates by selecting control individuals within the propensity score threshold set by the experimental group, can effectively reduce the estimation bias caused by too large propensity score difference if the matching radius is set strictly enough. This type of matching can realize efficient matching when the sample size is large, but the setting of the threshold needs several attempts to avoid improperly setting the matching failure, or the matching accuracy is not high. (3) Kernel matching, which operates by using the kernel function to weight multiple control group individuals, can effectively reduce the influence of extreme values and avoid relying on a single matching pair, thus improving the stability of estimation. This type of matching is more suitable for the case where the propensity score distributions of the experimental and control groups overlap more. However, this approach is more sensitive to the choice of bandwidth and may also result in poor matching quality. In conclusion, different algorithms for propensity score matching estimation have their own advantages, and in order to avoid the bias of the results brought by a single algorithm, this study presents the results under the three estimation methods as a way to partially reduce the doubts about the results.

(12)Furthermore, the tables have been presented as pictures/images in ./tiff format. However, it is the PLOS principle that tables should be presented as editable tables. Please revise accordingly.

Thanks to the editor for the reminder, and we have resubmitted the forms available for editing as requested.

---

## [Editor Report · Decision Letter 1]

4 Apr 2025

Exploring the Transmission Mechanism of Self discrepancy on Perceived Academic Stress--Based on the methods of posting grades ranking

PONE-D-24-06473R1

Dear Dr. Yi,

We’re pleased to inform you that your manuscript has been judged scientifically suitable for publication and will be formally accepted for publication once it meets all outstanding technical requirements.

Kind regards,

Ioannis G. Katsantonis, PhD, MPhil

Academic Editor

PLOS ONE

Additional Editor Comments (optional):

The authors have devoted a significant effort in addressing my comments and the concerns raised by the reviewer. Having carefully reviewed the paper, I believe that it can be published in its current form. During the proofreading process, I advise the authors to input all tables (tables 1 to 9) inside the main text
---

## [Editor Report · Acceptance letter]

PONE-D-24-06473R1

PLOS ONE

Dear Dr. Yi,

I'm pleased to inform you that your manuscript has been deemed suitable for publication in PLOS ONE. Congratulations! Your manuscript is now being handed over to our production team.

Kind regards,

on behalf of

Dr. Ioannis G. Katsantonis

Academic Editor

PLOS ONE